# Robust Approaches to the Quantitative Analysis of Genome Formula Variation in Multipartite and Segmented Viruses

**DOI:** 10.3390/v16020270

**Published:** 2024-02-08

**Authors:** Marcelle L. Johnson, Mark P. Zwart

**Affiliations:** Department of Microbial Ecology, The Netherlands Institute of Ecology (NIOO-KNAW), P.O. Box 50, 6700 AB Wageningen, The Netherlands; m.johnson@nioo.knaw.nl

**Keywords:** multipartite virus, segmented virus, genome formula, statistical analysis, RT-PCR, sequencing, plant virus, virus evolution, virus ecology

## Abstract

When viruses have segmented genomes, the set of frequencies describing the abundance of segments is called the genome formula. The genome formula is often unbalanced and highly variable for both segmented and multipartite viruses. A growing number of studies are quantifying the genome formula to measure its effects on infection and to consider its ecological and evolutionary implications. Different approaches have been reported for analyzing genome formula data, including qualitative description, applying standard statistical tests such as ANOVA, and customized analyses. However, these approaches have different shortcomings, and test assumptions are often unmet, potentially leading to erroneous conclusions. Here, we address these challenges, leading to a threefold contribution. First, we propose a simple metric for analyzing genome formula variation: the genome formula distance. We describe the properties of this metric and provide a framework for understanding metric values. Second, we explain how this metric can be applied for different purposes, including testing for genome-formula differences and comparing observations to a reference genome formula value. Third, we re-analyze published data to illustrate the applications and weigh the evidence for previous conclusions. Our re-analysis of published datasets confirms many previous results but also provides evidence that the genome formula can be carried over from the inoculum to the virus population in a host. The simple procedures we propose contribute to the robust and accessible analysis of genome-formula data.

## 1. Introduction

Many viruses have segmented genomes: their complete hereditary material consists of multiple nucleic acid molecules. Packaging these genome segments into virus particles can result in various distributions of genome segments over virus particles [1,2] (Figure 1). Segmented viruses package one copy of each genome segment into each virus particle (Figure 1b). This arrangement is thought to ensure genome integrity and maximize opportunities for virus transmission. By contrast, multipartite viruses package each genome segment into a separate virus particle (Figure 1c). This arrangement results in a dependence on multiple virus particles for successful virus transmission, and it is thought to make transmission less efficient and, thereby, impose a substantial cost to virus spread [3,4]. Interestingly, some viruses blur the distinction between segmented and multipartite viruses. These viruses do not always package a full complement of genome segments into each virus particle [5,6], resulting in transmission that depends partly on incomplete particles [7,8,9] (Figure 1d). Whereas segmented viruses are most common among animal viruses, multipartite viruses abound among plant viruses [2,10]. However, there are many examples of segmented plant viruses [2,10]. At least one multipartite animal virus has been identified [11], and there are likely more cases [2,12].

For some multipartite and segmented viruses, variation in the frequency of genome segments has been observed [8,13,14,15,16]. The genome formula is the abundance of all virus genome segments, and it is typically described in one of two ways. If we take a bi-segmented virus with segments at equal abundance as an example, the genome formula can be expressed as a ratio 1:1 (segment1:segment2) or as a set of relative frequencies {0.5, 0.5} {segment1, segment2}. We use the latter convention throughout this paper. Current interest in the genome formula was sparked by the seminal work of Sicard and coworkers on faba bean necrotic stunt virus (FBNSV), a multipartite DNA virus with eight genome segments [13]. These authors showed that the genome formula converges on an unbalanced equilibrium when disrupted, and this equilibrium is host-species-dependent. Notably, the authors also observed considerable variation within and between plants in the genome formula, highlighting its stochastic nature. Later work confirmed similar findings for alfalfa mosaic virus (AMV), a multipartite plant RNA virus with three genome segments [14]. From a historical perspective, it is interesting to note that previous observations already showed the variable nature of the genome formula for multipartite [17] and segmented [18,19] viruses, even if the implications may not have been acknowledged then. In the meantime, genome formula variation has also been shown for segmented animal viruses [8,16]. Although studies on the genome formula have focused on full-length virus genome segments [13,14,20], other genetic elements are also relevant. For example, many RNA viruses produce sub-genomic RNAs, and for some viruses, these RNAs can be packaged into virus particles [21]. Parasitic genetic elements such as satellites are also known to affect the genome formula [22,23], and a full understanding will, therefore, require considering these elements. Given that genome formula variation appears to be a feature of many virus–host systems, what are the causes and consequences of this variation?

Both random and directional forces are likely to shape variation in the genome formula. Population bottlenecks are likely to result in stochastic variation in the genome formula. When the total number of segments entering a cell is small, the frequencies of the different segment types are likely to vary, a process known as genome formula drift [24]. Sicard et al. (2013) suggested that variation in the genome formula is similar to copy number variation (CNV), possibly affecting gene expression and, thereby, enabling a rapid tuning of gene expression [13]. Under this hypothesis, selection for a beneficial genome formula would also be a directional force [25]. Other directional forces may include differences in the rates of replication or encapsidation for different segments [1,14].

Many plant viruses that cause disease and economic losses in cultivated plants are multipartite or segmented viruses, including viruses with very broad host ranges [26]. For example, the multipartite viruses cucumber mosaic virus (CMV) and AMV have broad host ranges, as does the segmented tomato spotted wilt virus (TSWV) [27]. Having three or four genome segments has been identified as a predictor for a large host range in plant viruses [28]. As genome formula changes may enable these broad host ranges [1], the genome formula may also have relevance for understanding virus emergence and disease outbreaks. There are no reports of genome formula variation in real-world virus populations; still, we speculate that the genome formula might have value as a tool for the monitoring of virus populations in crops and predicting disease outcomes. Finally, theory suggests that agro-ecosystems may also be conducive to the propagation of multipartite viruses due to many opportunities for transmission in dense monocultures [29]. For these reasons, studying the infection dynamics and genome formula variation of multipartite viruses in experiments and in agricultural ecosystems is a relevant topic within plant virus epidemiology.

**Figure 1 viruses-16-00270-f001:**
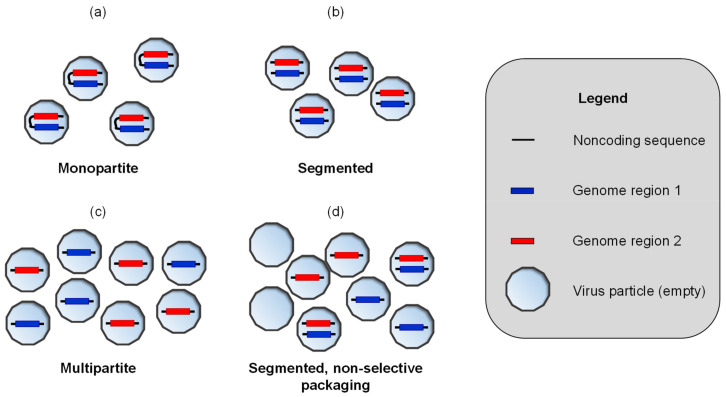
We provide a schematic illustration of the variation in the distribution of genome segments (nucleic acid molecules) over virus particles. A legend is given on the far right. In each case shown, we assume the virus genome consists of the two identical coding genome regions, identified by blue and red fills, forming one or two segments. (**a**) Monopartite viruses have a single genome segment. Note that the two genome regions form a single molecule in the illustration. (**b**) Segmented viruses have multiple genome segments: two genome segments in this example. These viruses package a full complement of genome segments into each virus particle. (**c**) A multipartite virus with two genome segments is shown. Each segment is packaged individually into a virus particle. Infection will depend on the transmission of multiple virus particles, as both a blue and a red segment are needed. (**d**) A segmented virus with non-selective packaging is shown. The illustration is a hypothetical distribution based only on the observation that for some segmented viruses, many virus particles have an incomplete set of genome segments [5,19]. This organization is included to highlight that many distributions of genome segments over virus particles are possible, and that the genome formula of segmented viruses does not have to be balanced (i.e., not 1:1 ratio of genome segments).

Most studies quantify the genome formula with the same molecular method. For DNA viruses, quantitative polymerase chain reaction (qPCR) is used, whereas RNA viruses require reverse transcription—qPCR (RT-qPCR). In these assays, specific primers are used to amplify distinct template sequences on the different genome segments, and SYBR-Green-induced fluorescence is used to quantify amplicon copy numbers. For those viruses that generate subgenomic RNAs, primers are designed to amplify templates that only occur in the full-length RNA [20]. One study compared three other methods to RT-qPCR for the quantification of the CMV genome formula: RT—digital droplet PCR (RT-dPCR), Illumina short-read sequencing, and Oxford Nanopore Technologies (ONT) long-read sequencing. This study found that the methods give roughly similar results, although there are systematic differences [20]. Another study on FBNSV showed that rolling circle amplification (RCA), a common amplification step before sequencing for circular DNA viruses, may lead to discrepancies in the quantification of the genome formula compared to qPCR [30].

Once the genome formula has been quantified, there are several different approaches for analyzing these data, driven in part by different research questions. For many studies, a key question is how to make rigorous genome formula comparisons for two or more groups. To show the breadth of approaches used to address this question, we provide a non-exhaustive overview (Table 1). When we consider the strengths and weaknesses of these approaches, we see that most approaches used have some crucial shortcomings (Table 1). In many cases, model assumptions are not met, or the procedure can only be applied to a bipartite virus or one specific genome segment. Ideally, we want a single method for comparing the complete genome formula with a limited set of model assumptions that can be met in practice.

While there are compelling hypotheses about the genome formula, exploring the causes and consequences of genome formula variation will require robust approaches. To date, studies have used a plethora of different approaches, ranging from simple qualitative comparisons to employing sophisticated statistical methods. This study is focused on these analysis methods and their effect on outcomes. Based on our previous experience with developing approaches for analyzing genome formula data, our hypothesis is that the method used can have a critical effect on study outcome. The result we work towards is having a robust, well-documented approach to analyzing genome formula data, which has been applied to various datasets, illustrating its applications and demonstrating its relevance. Here, we propose a simple and robust approach to genome formula analysis that relies on the genome formula distance metric [15]. We document this method in detail as a resource for the analysis of genome-formula data. We provide a framework for interpreting our metric’s values and explore how this approach can be applied to different problems. Finally, we re-analyze some previously published datasets to illustrate the benefits of this approach and as a validation of previous analyses.

## 2. Methods

All analyses were performed with R version 4.3.1 software for statistical computing [32]. Calculations of the genome formula distance were performed with the *vegdist* function, PERMANOVA was performed with the *adonis2* function, and PERMDISP2 was performed with the *betadisper* and *permutest* functions, which all pertain to the vegan Community Ecology Package version 2.6-4 [33].

All code for analysis and the data formatted for analysis are available as R markdown files at Zenodo (10.5281/zenodo.10355273). Access to the submission is currently restricted to avoid any confusion prior to the availability of the paper; please follow this link to gain access.

## 3. Results

### 3.1. The Genome Formula Distance Metric

Given the shortcomings of many methods for analyzing genome formula variation, we recently developed another approach, based on the genome formula distance metric [15], in combination with permutation-based statistical approaches [34,35]. Here, we build on this previous work by describing this metric in detail and considering some of its attributes, such as the range of values and its interpretation.

#### 3.1.1. The Genome Formula Distance Metric

We consider the genome formula (*G*) as the set of relative frequencies for all virus genome segments. For a viral genome with *k* segments:(1)$$G=\left\{f_{1},f_{2},\left[\ldots \right]f_{k}\right\}$$

Here, *f* is the relative mean frequency of a segment, such that for the *j^th^* segment:(2)$$f_{j}=c_{j}/\sum_{i=1}^{k}c_{i}$$

Here, *c* is a measurement of accumulation for a specific segment, such as quantitative polymerase chain reaction (qPCR) measurements. Per definition, the sum of all *f* values is 1. When any measurement of segment accumulation *c* changes, it will affect the relative frequency of all other segments.

To compare two values of the genome formula, in a previous study, we proposed to consider the Euclidean distance between them [15]. We refer to this metric as the genome formula distance (*D*), such that for two genome formula observations *a* and *b*, the distance between them is as follows:(3)$$D_{a,b}=\sqrt{\sum_{i=1}^{k}{\left(f_{a,i}-f_{b,i}\right)}^{2}\;}$$

Intuitively, *D* is simply the length of the straight line connecting two points in an *n*-dimensional space (Figure 2). The multivariate genome formula data are, therefore, reduced to a single distance value, simplifying analysis and removing the dependence between measurements expressed as relative frequencies. Although we previously described this metric and applied it for comparing groups of genome formula observations, we did not consider the properties of this metric in detail. Therefore, before considering here how this metric can be applied to data for several different goals, we describe some properties of this metric and generate expectations based on first principles in detail.

#### 3.1.2. Minimum and Maximum Values of the Genome Formula Distance Metric

Various properties of the metric *D* can be readily established. Its minimum value is $D_{a,b}^{min}=0$, which is when two genome formula values coincide. Its maximum value is $D_{a,b}^{max}=\sqrt{2}$, as can be shown by induction (Figure 2). For a bipartite virus, the greatest possible *D* will be obtained when $G_{a}=\left\{1,0\right\}$ and $G_{b}=\left\{1,0\right\}$, when $D_{a,b}=\sqrt{{\left(1-0\right)}^{2}+{\left(0-1\right)}^{2}}=\sqrt{2}$. For tripartite and tetrapartite, the greatest distance occurs along the edges of the genome formula space. These edges represent the line connecting *G* values composed of the presence of only one segment, resulting in $D_{a,b}=\sqrt{2}$ (Figure 2). In real life, we do not expect to see such large values, as we do not expect to see replicating virus populations in which only a single segment is present. Although it is possible for some multipartite viruses to lose and reacquire a segment [36], all or a number of core segments are often required for replication [3,37]. It is, therefore, interesting to consider what values of *D* can be expected under scenarios with a higher biological relevance.

#### 3.1.3. Distance Metric for Random Genome Formula Variation

To determine a plausible upper limit for the mean distance between two observations of the genome formula (${\bar{D}}_{a,b}$), we assume that all genome segments must be present in the virus population, but that the level of accumulation is, otherwise, entirely random. For each segment, we, therefore, sample a value from a uniform distribution and then determine the mean pairwise distance ${\bar{D}}_{a,b}^{rand}$. The values of ${\bar{D}}_{a,b}^{rand}$ depend on the number of genome segments, with a maximum value of 0.391 for a tri-segmented virus (Table 2). If we find similar values for ${\bar{D}}_{a,b}$ for a real-world virus population, this result would suggest a genome formula shaped by random levels of accumulation for the different segments.

#### 3.1.4. Distance Metric for Maximum Genome Formula Drift

Whereas the strength of genetic drift decreases monotonically as effective population size increases, the strength of genome formula drift is maximized at an intermediate effective population size [25]. Therefore, to determine the maximum level of genome formula drift that a single population bottleneck event can induce, we have to consider a range of bottleneck sizes. We assume that the total number of virus particles that initiates an infection follows a Poisson distribution with a mean value λ and consider the predicted genome formula distance over a broad range of λ values for different numbers of genome segments (Figure 3). The maximum genome formula distance values, ${\bar{D}}_{a,b}^{drift}$, are given in Table 2. As expected, these values are lower than those obtained for random genome formula variation (${\bar{D}}_{a,b}^{rand}$), as the assumption of a Poisson-distributed number of founders constrains the variation in genome segment frequencies. If a population shows similar values of ${\bar{D}}_{a,b}$, this suggests that the genome formula variation observed is equivalent to the maximum variation that can be generated by a single bottlenecking event.

### 3.2. Applications of the Genome Formula Distance Metric

To illustrate how this metric can be applied to experimental data, we re-analyze datasets from several studies on plant multipartite viruses. We do not attempt to reproduce all analyses in these original studies here. Rather, we focus on a few cases to illustrate how an approach based on the genome formula distance can be used. Note that all the genome formula data re-analyzed throughout this study were obtained through qPCR or RT-qPCR. The only exception is the methods comparison by Boezen and coworkers [20]. Here, for that study, we also explicitly address the effect of different methods on genome formula quantification, as was performed in the original work.

#### 3.2.1. Comparison of the Genome Formula to Theoretical Values

We defined clear expectations for the upper limit of the genome formula distance metric for the random accumulation of genome segments (${\bar{D}}_{a,b}^{rand}$) or the maximum amount of genome formula drift generated by a single population bottleneck (${\bar{D}}_{a,b}^{drift}$) (Table 2). First, we compare these theoretical predictions to observed values of genome formula distance (${\bar{D}}_{a,b}$). We obtain these observed values by re-analyzing genome formula data reported in three experimental studies in which the genome formula was measured in single leaves or whole plants [13,14,15]. For the tripartite RNA viruses AMV and CMV, we find that the observed values for the genome formula distance are below both of our reference values (Table 3), as expected for systems that appear to converge on an equilibrium value. Two out of three measurements for AMV are close to the value measured for CMV (~0.20), which is near to prediction for maximum genome formula drift (${\bar{D}}_{a,b}^{drift}\sim \;0.28$ for a tri-segmented virus). For the octapartite DNA virus FBNSV, we see a decrease in ${\bar{D}}_{a,b}$, indicating a reduction in variability over leaf levels (Table 3) as reported in the original study in Figure 3A [13]. The decrease in $D_{a,b}$ over leaf levels is highly significant (Kendall rank correlation: τ = 0.368, *N =* 77, *p* < 0.001). When we compare values of ${\bar{D}}_{a,b}$ to model predictions, we find that it is higher than ${\bar{D}}_{a,b}^{rand}$ in the inoculated leaf (leaf level 1) but falls to and remains at levels below the ${\bar{D}}_{a,b}^{drift}$ predictions by leaf level 3 (Table 3).

Overall, these comparisons between model predictions and observed values of ${\bar{D}}_{a,b}$ underscore that there is considerable genome formula variation, suggesting that stochastic forces play an important role in shaping the genome formula. The differences in variability for the AMV estimates might reflect differences between the inoculated and systemic leaves but may reflect the relatively low number of replicates for each condition (*n* = 6). This variability stresses the need for high levels of replication for the representative estimates of these indexes. For FBNSV, the higher-than-expected genome formula variation in the inoculated leaf is striking. However, this phenomenon is probably related to the inoculation with *Agrobacterium*, as once the virus has systemically moved, it no longer surpasses model predictions of ${\bar{D}}_{a,b}^{rand}$.

#### 3.2.2. Comparison of the Genome Formula for Different Groups

Boezen and coworkers first applied the genome formula distance metric to compare the genome formula for different treatments [15]. In this section, we first describe these previous results in detail, as they are important for understanding this approach and its limitations. This previous study explored the effects of mixed infection with other plant viruses on CMV’s genome formula [15]. To compare the genome formula of CMV in different treatments, the authors calculated the genome formula distances and then performed PERMANOVA. PERMANOVA is a permutational multivariate analysis of variance, a non-parametric ANOVA widely applied in ecology [34,35]. PERMANOVA is often applied to such analyses because of its robustness: the test makes fewer assumptions than parametric procedures. Note that if we apply PERMANOVA to the genome formula distance as suggested here, we are performing a univariate analysis, for which PERMANOVA is also suitable. One interesting feature of PERMANOVA is that the procedure detects both differences in mean (or centroid for multivariate data) and spread. If we detect a significant difference, we must rule out a significant difference in spread before we can conclude that there are differences in the mean. The PERMDISP2 procedure tests whether there are significant differences in spread [34]. When Boezen and coworkers applied this procedure, they found a significant difference between the PERMANOVA and PERMDISP2 procedures [15]. Therefore, in this case, the authors could only conclude that mixed infections had a significant effect on genome formula spread, surprisingly leading to a reduction in the spread compared to a CMV-only infection. Now that we have described this procedure and its application in previous work in detail, we consider how it can be applied to other datasets.

To further illustrate how PERMANOVA on the genome formula distance is useful, we re-analyzed data from four other experiments (see Appendix A for a detailed description). For the first dataset we consider here, the original study measured the genome formula of CMV with four different methods in three hosts [20]. The study found no effect of host species on the genome formula, and although the different methods gave similar results, there was a significant effect of method on the measured genome formula [20]. When we re-analyzed these genome formula data, we found largely similar results when comparing our new procedure to the model selection in the original study. The PERMANOVA-based procedure is more robust (Table 1) but still manages to identify some subtle species effects on the genome formula that were not detected by the original analysis (see Appendix A). The second dataset we considered was from a study that showed frequency-dependent selection results in an equilibrium for AMV’s genome formula, and it showed that the genome formula of this RNA virus is host-species-dependent [14]. A number of datasets are reported in this paper, and we choose to focus on one specific question for our re-analysis: are there differences in the genome formula in the inoculated leaf, for leaves inoculated with different genome formulae? Here, we did not find a significant effect (Appendix A). This result contradicts the result of the statistical test in the original study. However, all plant tissues were jointly analyzed in the original paper, whereas here, we focused exclusively on the inoculated leaf. From a biological perspective, it makes the most sense to look for an effect of the inoculum early in the infection process. In the final section of the results (Section 3.2.3), we explore a different approach to analyzing these AMV data that sheds more light on the underlying processes.

Next, we compared the genome formula distance for two sets of experiments on the octapartite FBNSV in a seminal study that reignited interest in these viruses [13]. The third dataset we re-analyzed considers the genome formula in different leaf levels [13], the same dataset we used to determine the pairwise distance between genome formula measurements (Table 3). As we found large differences in genome formula variability (Table 3), we expect and indeed find that the PERMDISP2 result is significant (Appendix A). The results of the distance measurements and PERMANOVA are in good agreement. The original study used ANOVA to analyze the coefficient of variation for the genome formula in different leaf levels, also finding significant differences in variation between leaf levels [13]. Second, we considered the FBNSV genome formula in two plant species [13], for which the authors analyzed the abundance of individual segments. In agreement with the original analyses, we find highly significant differences in the genome formula distance between the two plant species, while the experiments in the same plant species render similar results (Appendix A).

These examples illustrate how readily our proposed approach can be used to analyze genome formula data. Our results are largely congruent with previous results in three out of four cases. However, there is a discrepancy for the data of Wu et al. on AMV infection [14], for which we analyzed a subset of the data using a different approach. This discrepancy illustrates that the approach and methods used matter for the results obtained.

#### 3.2.3. Comparison of the Genome Formula to Reference

We can also use the genome formula distance metric to compare observations of the genome formula to a reference. The reference genome formula used will depend on the question being addressed. We provide some examples to illustrate a range of reference values and a purpose for the comparison, to show the breadth of potential applications. These possible reference values include the following: (i) the mean genome formula for a group of observations (which, in effect, also occurs for PERMANOVA); (ii) the genome formula used in the inoculum for an experiment, to test whether it is maintained; (iii) a balanced genome formula (i.e., 1:1:1), to quantify the imbalance in the genome formula (see examples using another metric [13,16]); or (iv) theoretical predictions of the genome formula, to fit models to data and test these predictions. One example from previous work is worthy of mention because the authors used what is effectively the same metric we are proposing: Wu and coworkers used the genome formula distance metric to consider whether there was higher virus accumulation as virus populations approached the mean genome formula value [14]. A rank correlation was used to test for an association between genome formula distance and accumulation, and the results were significant. Now that we have given some examples of purposes for which reference values can be used in combination with our metric, next we consider one application in detail.

We previously considered whether there were significant differences for the AMV genome formula measured in inoculated leaves [14] when the inoculum genome formula is considered for the treatment (see Section 3.2.2 and Appendix A). However, in this instance, one could ask a more specific question: is the genome formula measured in the inoculated leaf more similar to the genome formula of the inoculum than expected by chance? To address this question, we first calculate the mean genome formula distance for each AMV observation to its corresponding inoculum [14]. Next, we resampled the data by randomly assigning observations to inocula and calculated the mean genome formula distance for a large number of resampled datasets (10^4^). We can then compare the observed outcome to the predicted range of genome formula distances for the resampled data to determine its likelihood. This analysis clearly shows that the observed genome formula distance is less than that predicted for the resampled data, showing that there is a clear effect of the inoculum on the genome formula measured in the inoculated leaf (Figure 4, Table 4). The genome formula distance is much smaller than the predicted value for randomized data, showing that the inoculum has a clear effect on the genome formula.

This result appears to contradict the PERMANOVA test results on the same data, in which there was not a significant treatment effect. However, these two procedures address different questions and test different null hypotheses. Rather than considering whether there is an effect of treatment on the mean, here, we are asking whether means are closer to a reference corresponding to each treatment. The resampling test we have used in this section incorporates more information from the experimental setup, resulting in a specific null hypothesis that can be more readily rejected.

Finally, we can perform the same resampling procedure for other tissues analyzed in the same experiment, in which case we do not see an effect in any other tissue (Table 4 and Appendix B). Therefore, the effect of the inoculum on the genome formula appears to be transient, as this effect is absent in systemically infected tissues. In summary, by reanalyzing these data, we do find strong evidence for an effect of the inoculum: in the inoculated leaf alone, the genome formula is closer to the inoculum genome formula than would be expected by chance.

## 4. Discussion

In the past decade, there has been considerable interest in the genome formula of both multipartite and segmented viruses [1,2,8,13,14,25,36,38]. However, different studies have applied different analysis methods, many of which have serious shortcomings. To address this challenge and provide examples, here, we present some simple and robust approaches to analyzing genome formula data. Our approach is based on the genome formula distance metric, the Euclidean distance between two genome formula values. We demonstrated the properties of this metric and showed how it can be applied to different analyses. By reanalyzing previously published datasets, we showed that in some cases, the approach used matters for the outcome, in support of our expectation. The genome formula distance is amenable to formal analysis by simple and robust approaches such as PERMANOVA, using existing software packages such as the vegan package for community ecology in R [33].

We argue that permutational analyses based on the genome formula distance are superior to other approaches used to analyze genome formulae, primarily because the assumptions of the statistical test are met with this procedure. Many of the procedures used previously by others and ourselves do not meet these assumptions, with one common violation being the assumption of independence when relative frequencies are analyzed as independent measurements. The procedures we propose here avoid this problem by reducing relative frequencies to a single distance measurement. Ultimately, the main benefit of the procedures we are proposing is greater robustness and, consequently, validity, irrespective of test performance. Nevertheless, in two cases, this procedure found differences where other procedures did not find any, suggesting that the statistical power of these procedures is not lower.

Most of our reanalysis yielded similar results to the original study. For the work of Wu and coworkers [14], our initial re-analysis of the inoculated leaf contradicts the study’s results, whereas our subsequent re-sampling analysis determined a clear effect of the inoculum on the genome formula in the inoculated leaf. By inference, there are, therefore, some differences between plants due to the inoculum, in agreement with the studies’ conclusions. The different test results for the PERMANOVA and re-sampling based approaches are logically compatible given the different null hypotheses being evaluated, and they illustrate the importance of carefully considering which hypothesis to test. Ultimately, the results convincingly show a clear legacy of the inoculum genome formula in the inoculated leaf. What could explain this outcome? It cannot be categorically ruled out that the in vitro synthesized inoculum has an effect, although this is highly unlikely given the instability of RNA under ambient conditions. The most likely explanation is, therefore, that insufficient generations of virus replication occurred for a frequency-dependent selection to alter the genome formula. Major changes in the genome formula might also be more likely to occur upon systemic movements of multipartite viruses, especially if these are associated with low multiplicities of cellular infection (MOI) that are predicted to facilitate rapid changes when using a theoretical model [25]. What is exciting about this new result is that it shows that the genome formula can be transmissible, as this is an essential ingredient for its hypothesized role in virus adaptation to changing host environments [1,13,25].

### 4.1. Alternative Metrics for Analyzing Genome Formula Data

In their landmark study on the FBNSV genome formula, Sicard et al. and coworkers [13] proposed Δ*GF* as a metric, which is expressed in general terms as follows:(4)$$\Delta {GF}_{a,b}=\sum_{i=1}^{k}\left|f_{a,i}-f_{b,i}\right|/2$$

This metric has been used for quantifying the imbalance in the genome formula (e.g., comparing empirical values to a balanced genome formula) [13,16]. Given that we advocate reducing multivariate data to a single distance measurement and then using permutational statistics, Δ*GF* also could be used instead of the genome formula distance *D* and often yield similar results. We chose the genome formula distance metric mainly because it provides the simplest and most intuitive representation of the distance between two data points in an *n*-dimensional space, i.e., a straight line. Another advantage may be that squaring differences will more heavily weigh larger distances. Ultimately, both approaches are reasonable, and the effect on the results of analysis may often be small. To facilitate the interpretation of analyses based on the Δ*GF* metric, we also calculated expected values of genome formula variation for a random accumulation of segments and under maximum genome formula drift (Appendix C).

### 4.2. Caveats

The approaches we propose have some important benefits, but it is important to keep in mind some limitations. First, when samples have significant differences in genome formula spread (i.e., as indicated by the PERMDISP2 procedure), no firm conclusions can be reached on differences in mean using PERMANOVA. Significant differences in spread between treatments can also be interesting in their own right. For example, Boezen and co-workers used this procedure to show that mixed infections restricted genome formula variation [15]. However, if there is not a framework to interpret whether differences in spread are relevant, this outcome may not be very informative. Second, in some cases multipartite viruses can lose or gain genome segments that are not essential for replication [36]. The approaches we propose can handle such data, as segments can have a relative frequency of zero. However, when segments are missing altogether, we suggest considering other approaches for analysis. For example, essential FBNSV segments (e.g., R and S) are typically present at low frequencies (f<0.05). Their complete absence would have a minimal effect on the hypothetical GF distance, but result in virus populations incapable of replication. Third, methods used for the quantification of the genome formula can have an effect on the results, as shown previously [20] and confirmed by our re-analysis here (Appendix A). The analysis of results obtained with different methods clearly should be avoided. However, as the genome formula quantification method could induce different amounts of technical variation, a comparison of indexes like genome formula distance (${\bar{D}}_{a,b}$) obtained with different methods should also be avoided.

### 4.3. Concluding Remarks

Genome formula data can have a large number of dimensions, complicating their visualization, analysis, and, ultimately, the interpretation of results. The visualization of these data can be aided with the use of ternary plots or radar charts, whereas, here, we explore new approaches to the analysis. We show that the genome formula distance metric can be used for a number of different purposes, ranging from comparisons between experimental treatments to comparing data and theoretical expectations. One major advantage of these approaches is their simplicity and reliance on well-established statistical tests, such as PERMANOVA. However, other developments suggest future directions for analyzing these kinds of datasets. First, ecological communities, such as microbiomes, often have high species richness. Advanced approaches for analyzing the relative frequency of taxonomic units [39] could serve as inspiration for how to refine methods for genome formula analysis. Second, machine learning and deep learning algorithms [40] may prove to be valuable for analyzing genome formula data, as these tools may identify trends that are difficult to visualize and may not be identified by testing hypotheses specified a priori.

## Figures and Tables

**Figure 2 viruses-16-00270-f002:**
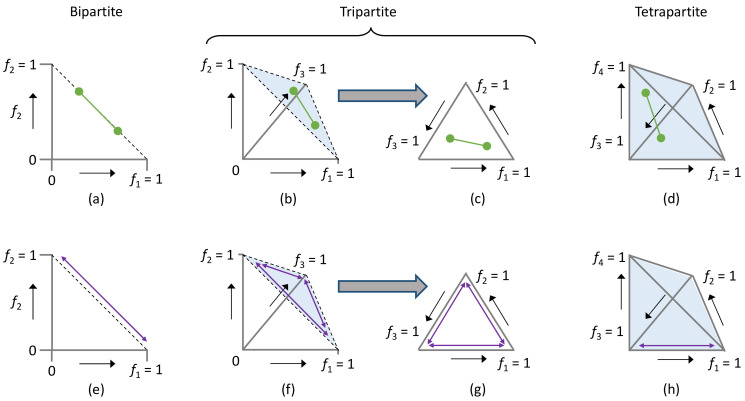
Here, we illustrate the genome formula distance metric (**top** panels, green lines) and its maximum possible distance for different numbers of genome segments (**bottom** panels, purple arrows). Figure axes are genome segment frequencies (*f*) for 2 (panels (**a**,**b**)), 3 (panels (**b**,**c**,**f**,**g**)), or 4 genome segments (panels (**d**,**h**)). (**a**) For a bipartite virus, we illustrate two possible genome formula values with green points and the distance between them with a line. Note that for the bipartite virus, all possible genome formula values fall on the dotted line connecting (1,0) and (0,1). (**b**) For a tripartite virus, we illustrate two possible genome formula values in three-dimensional genome formula space. As the sum of relative frequencies is 1, all possible genome formula values fall in the triangular plane illustrated by the dotted lines and light blue shading. (**c**) As all values fall in the same plane in panel b, genome formula values for a tri-segmented virus are often illustrated in only this plane, resulting in a ternary plot. (**d**) Two genome formula values and their distance are illustrated for a tetrapartite virus in a quarternary plot. All values in the tetrahedron represent possible genome formula values, as indicated by the light blue shading. (**e**) The maximum possible genome formula distance for a bipartite virus is simply the line connecting the points (1,0) and (0,1). (**f**) For the tripartite virus, the longest possible distance in the genome formula space is attained along its borders, resulting in an identical maximum genome formula distance to the bipartite virus. The light blue shading indicates the possible space for genome formula values. (**g**) The outcome described in panel f is clearer in the ternary plot of the genome formula space. (**h**) For a tetrapartite virus, there is no distance between two points in the genome formula space that is longer than the maximum distance for the bipartite and tripartite viruses. This maximum distance occurs at the edges of the genome formula space, as indicated by the light blue shading, connecting the vertices, which represent the presence of a single segment. To keep the panel clear, we only illustrate this for one edge for a tetrapartite virus, although there are six such edges.

**Figure 3 viruses-16-00270-f003:**
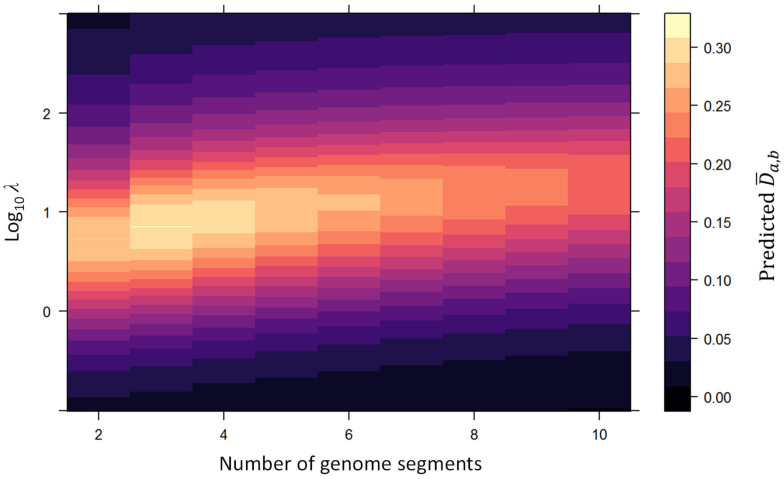
The effects of the number of segments and bottleneck size on the predicted genome formula distance are illustrated. The *x*-axis indicates the number of virus genome segments, whereas the *y*-axis indicates the log-transformed number of infection founders (*λ*). For all combinations of these values, we predicted the mean genome formula distance ${\bar{D}}_{a,b}$, a value indicated by the heat according to the legend on the far right. We used these simulation results to determine the highest value of ${\bar{D}}_{a,b}$ for each number of genome segments, a value we term ${\bar{D}}_{a,b}^{drift}$. Note that the highest mean distance values occur at intermediate values of *λ*, as well as being associated with higher values of *λ* as the number of segments is increased.

**Figure 4 viruses-16-00270-f004:**
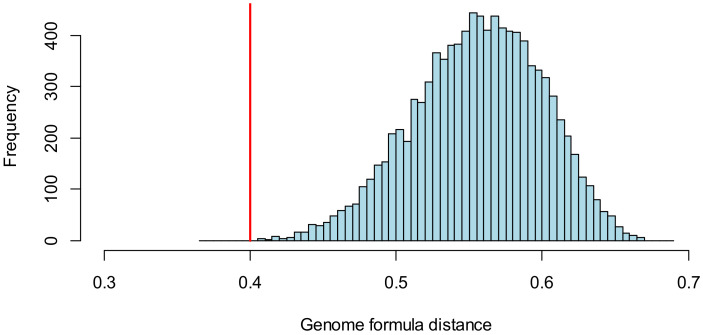
Resampling approach to testing for an effect of inoculum on the genome formula measured in the inoculated leaf. The blue bars in the histogram indicate the frequency of predicted mean genome formula distance for 10^4^ resampled datasets, in which observations in the inoculated leaf were randomly assigned to an inoculum. The red line indicates the genome formula distance for the actual data.

**Table 1 viruses-16-00270-t001:** Approaches to comparing genome formula values for two or more groups.

Approach	Strengths	Weaknesses	Ref.
Analysis of variance (ANOVA) on the relative frequencies of individual genome segments	(i) Parsimony of the analysis	(i) Limited to the analysis of individual segments (ii) Model assumptions ^1^	[13]
Multivariate analysis of variance (MANOVA) on the relative frequency of all genome segments	(i) Single analysis of all segments (ii) Technical error included in the analysis	(i) Dependence between relative frequencies (ii) Model assumptions ^1,2^	[14]
Model selection based on the ΔGF metric ^3^ for all genome segments	(i) Single analysis of all segments	(i) Assumptions for estimating the likelihoods and weighing of model parameters for model selection ^4^	[20]
T-tests on ratio of the log-tranformed RNA1:RNA2	(i) Parsimony (ii) Model assumptions met	(i) Only applicable to bipartite viruses (ii) Consider effects of a single factor	[31]
PERMANOVA on the genome formula distance metric ^5^	(i) Parsimony (ii) Single analysis of all segments (iii) Model assumptions met	(i) If there are differences in spread, differences in centroid cannot be assessed	[15]

^1^ Normality of the residuals and equality of variance assumptions may not be met. For the comparison of single segments with ANOVA, the assumption of independence of observations is met. For comparison of multiple segments, the assumption is violated. ^2^ In addition to ANOVA assumptions, MANOVA assumes no multivariate outliers. ^3^ The cumulative distance between genome formula observations and a reference value [13], which, in this case, is the mean value for the group under consideration. ^4^ To calculate the negative log likelihood for these data, residuals are assumed to be normally distributed. In addition, each group mean is weighed as a free parameter for model selection, whereas it follows directly from the data. ^5^ This metric is described in detail in Section 3.2.

**Table 2 viruses-16-00270-t002:** Expected values of *D* for random genome formula variation (${\bar{D}}_{a,b}^{rand}$) or the maximum genome formula drift introduced by a single bottleneck event (${\bar{D}}_{a,b}^{drift}$).

Number of Genome Segments	${\bar{D}}_{a,b}^{rand}$	${\bar{D}}_{a,b}^{drift}$	*λ* ^1^
2	0.3855	0.2877	5.37
3	0.3905	0.2801	7.08
4	0.3638	0.2629	9.12
5	0.3367	0.2494	10.47
6	0.3132	0.2341	12.30
7	0.2934	0.2189	14.12
8	0.2767	0.2060	15.85
9	0.2625	0.1929	18.20
10	0.2501	0.1847	19.50

^1^ The bottleneck value corresponding to the maximum ${\bar{D}}_{a,b}^{drift}$ value.

**Table 3 viruses-16-00270-t003:** Observed values for the genome formula distance (${\bar{D}}_{a,b}$) for two tripartite viruses.

Genome Segments	Model Predictions ^1^					
	${\bar{D}}_{a,b}^{rand}$	${\bar{D}}_{a,b}^{drift}$	Ref	Experiment	n	${\bar{D}}_{a,b}$ ± SD
3	0.391	0.280	[14]	AMV in *N. benthamiana*, inoculated	6	0.077 ± 0.015
				AMV in *N. benthamiana*, lower leaf	6	0.195 ± 0.029
				AMV in *N. benthamiana*, upper leaf	6	0.197 ± 0.124
			[15]	CMV in *N. tabacum*, whole plant	9	0.207 ± 0.069
8	0.277	0.206	[13]	FBNSV in *V. faba*, leaf level 1	9	0.352 ± 0.097
				FBNSV in *V. faba*, leaf level 2	8	0.275 ± 0.062
				FBNSV in *V. faba*, leaf level 3	13	0.198 ± 0.045
				FBNSV in *V. faba*, leaf level 4	15	0.175 ± 0.050
				FBNSV in *V. faba*, leaf level 5	16	0.198 ± 0.063
				FBNSV in *V. faba*, leaf level 6	16	0.178 ± 0.031

^1^ Predictions of the mean genome formula distance under random accumulation (${\bar{D}}_{a,b}^{rand}$) and the maximum genome formula drift introduced by a single bottleneck event (${\bar{D}}_{a,b}^{drift}$) are given, depending on the number of genome segments, as given in Table 2.

**Table 4 viruses-16-00270-t004:** Re-analysis of the AMV genome formula data [14] with a resampling approach.

	Genome Formula Distance to Inoculum		
Tissue	Observed ^1^	Predicted ^2^	Ranking ^3^
Inoculated leaf	0.400 ± 0.242	0.556 [0.434–0.652]	5
Middle leaf	0.484 ± 0.261	0.494 [0.410–0.568]	3683
Upper leaf	0.530 ± 0.237	0.503 [0.418–0.576]	7919
Rest of plant	0.445 ± 0.245	0.486 [0.421–0.538]	533

^1^ The observed value of the mean genome formula distance to the inoculum in the corresponding tissue, with its standard deviation. ^2^ The predicted value of the mean genome formula distance based on randomized datasets, with its 99% confidence interval. ^3^ The number of randomized datasets for which the mean genome formula distance was smaller than the observed value, out of 10^4^ resampled datasets in total. Ranks < 250 or > 9750 fall outside of the 95% confidence interval, while ranks <50 or >9950 fall outside of the 99% confidence interval.

## Data Availability

No new data were generated in this study. All code and the datasets re-analyzed are available at Zenodo (10.5281/zenodo.10355273).

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
