# Peer review of "Robust Approaches to the Quantitative Analysis of Genome Formula Variation in Multipartite and Segmented Viruses"

_viruses, 2024, doi:10.3390/v16020270_

Round 1

Reviewer 1 Report

Comments and Suggestions for Authors

The work by Johnson and Zwart entitled “Robust approaches for the quantitative analysis of genome formula variation in multipartite and segmented viruses” innovated in the customization and testing of a genome formula distance using published data. Overall, the note is well written and highlights avenues for future studies. However, before recommending acceptance, I have the following suggestions, specially concerning analytics.

First, authors should explicitly mention a research hypothesis to be tested, as well as the expected results, both in the last paragraph of the introduction after the research gap (L80) and the study goal (L83). This will allow readers focusing on explicit expectations when approaching the technical note, which ideally must be hypothesis-driven as compared to only descriptive.

Second, authors should improve Figure 2 (L200) so that the grids do not appear pixelated. A higher grid resolution, whenever possible, would be desirable to improve its readability and contrast.

Third, the discussion is perceptive, although slightly short (perhaps OK for a technical note). Still, I would encourage authors to envision a closing paragraph at the end of the discussion (in L403) containing the study caveats and their corresponding recommendations.

Last but not least, please close the discussion section with two new sub-heading, one in L404 on perspectives, and another one in L405 with concluding remarks.

Author Response

We thank the reviewers for evaluating the manuscript and for their constructive comments. Below, reviewer comments are reproduced and our responses are given. Line numbers refer to the revised manuscript, in which changes have been highlighted in blue.

Comment: The work by Johnson and Zwart entitled “Robust approaches for the quantitative analysis of genome formula variation in multipartite and segmented viruses” innovated in the customization and testing of a genome formula distance using published data. Overall, the note is well written and highlights avenues for future studies. However, before recommending acceptance, I have the following suggestions, specially concerning analytics.

 Response: We thank the reviewer for their appreciation of our work.

Comment: First, authors should explicitly mention a research hypothesis to be tested, as well as the expected results, both in the last paragraph of the introduction after the research gap (L80) and the study goal (L83). This will allow readers focusing on explicit expectations when approaching the technical note, which ideally must be hypothesis-driven as compared to only descriptive.

 Response: We thank the reviewer for this helpful suggestion. Although the primary focus is on improving quantitative analyses, it is indeed possible to formulate and evaluate hypotheses pertaining to or stemming from this analysis. We have now included a hypothesis and an expectation in the introduction (lines 138-144).

Comment: Second, authors should improve Figure 2 (L200) so that the grids do not appear pixelated. A higher grid resolution, whenever possible, would be desirable to improve its readability and contrast.

 Response: We thank the reviewer for this suggestion. Although we agree that smoothing would lead to an aesthetically more pleasing figure, the pixilation in the image is intentional. The main reason for this decision is that the number of genome segments (x-axis) is a discreet quantity, and by not having a smooth contour, the graph emphasizes that predictions can only be made for integral numbers. An additional benefit is that the step size for Log Lambda is directly apparent from the graph. Finally, the color combination chosen has been selected for the contrast between shades and accessibility for the visually impaired.

Comment: Third, the discussion is perceptive, although slightly short (perhaps OK for a technical note). Still, I would encourage authors to envision a closing paragraph at the end of the discussion (in L403) containing the study caveats and their corresponding recommendations.

Last but not least, please close the discussion section with two new sub-heading, one in L404 on perspectives, and another one in L405 with concluding remarks.

 Response: We thank the reviewer for this helpful suggestion. We have revised the discussion accordingly. We restructured it so that the most obvious discussion points come first (lines 404-460), followed by the discussion of the alternative metric already present in the previous version (lines 462-476), and new sections on caveats (lines 478-497) and concluding remarks, which includes some perspectives for future developments (lines 499-512). These final three parts have subheading to distinguish them, as suggested by the reviewer.

Reviewer 2 Report

Comments and Suggestions for Authors

The presented manuscript entitled ´Robust approaches for the quantitative analysis of genome formula variation in multipartite and segmented viruses´, classified as Technical Note, focuses on the statistical approaches used for the analysis of the genome formula of viruses with segmented genomes and their differences in infected plants.

The authors summarize the pros and cons of the methods used in the previously published works, some of which they are co-authors, and formulate the proposed distance metric approach. They define the theoretical values of distance metrics, re-analyse the published real data of qPCR detected concentration of AMV, CMV and FBNSV viral segments and their genome formulas.

The Introduction is clear, summarises all the important information, chapter Methods refers to the R analyses performed but it is pity that the

The ´Results´ describes in detail theoretical distance metrics – proposed formula and value for viruses with segmented genomes; compares them with the selected references.

My comments relate more to the formal presentation, to the grasp of the text and its přehlednost to the reader.

-          Despite the theoretical nature of the work presented in the manuscript, the chosen structure of the Results chapter is a bit confusing, as some parts of the approaches used  are explained in detail, including well-known principles, and some of the data obtained here are directly compared (and discussed) with the originally published conclusions /for example part L244-261, L294-303, etc./.

I consider this to be the weakest part of the manuscript.

I recommend to consider reformulating of the Results and splitting the results and discussion sections in favour of Discussion.

-          I don´t see any reason for the part 3.1 – i.e. Overview of approaches - to be included in the Result, I recommend moving this part to the Introduction.

-          I do not consider it happy when it is necessary to work with references (other published papers) to understand the results, a large part of whichis presented in the Appendices, and when the original published results are referred to in the form of “Fig 3A in the original study...”

-          The numbering of Results Sections in the text should be corrected (L275, L307, etc.).

Author Response

We thank the reviewers for evaluating the manuscript and for their constructive comments. Below, reviewer comments are reproduced and our responses are given. Line numbers refer to the revised manuscript, in which changes have been highlighted in blue.

Comment: The presented manuscript entitled ´Robust approaches for the quantitative analysis of genome formula variation in multipartite and segmented viruses´, classified as Technical Note, focuses on the statistical approaches used for the analysis of the genome formula of viruses with segmented genomes and their differences in infected plants.

The authors summarize the pros and cons of the methods used in the previously published works, some of which they are co-authors, and formulate the proposed distance metric approach. They define the theoretical values of distance metrics, re-analyse the published real data of qPCR detected concentration of AMV, CMV and FBNSV viral segments and their genome formulas.

The Introduction is clear, summarises all the important information, chapter Methods refers to the R analyses performed but it is pity that the

The ´Results´ describes in detail theoretical distance metrics – proposed formula and value for viruses with segmented genomes; compares them with the selected references.

My comments relate more to the formal presentation, to the grasp of the text and its přehlednost to the reader.

Response: We thank the reviewer for their appraisal of our work.

Comment: Despite the theoretical nature of the work presented in the manuscript, the chosen structure of the Results chapter is a bit confusing, as some parts of the approaches used  are explained in detail, including well-known principles, and some of the data obtained here are directly compared (and discussed) with the originally published conclusions /for example part L244-261, L294-303, etc./.

I consider this to be the weakest part of the manuscript.

Comment: I recommend to consider reformulating of the Results and splitting the results and discussion sections in favour of Discussion.

Response: We understand that the distinction between previous work and our work here may not have been clear in some parts of the results section. To remedy this problem, we have chosen to add additional text to further emphasize the difference between old and new results throughout the results section (e.g., lines 164-166, 182-184, 254-258, 295-298, 313-315, 372-374). If we would strictly separate results and discussion, we feel the results section would not be accessible and the discussion would become inordinately long and confusing. One important advantage of the way we have structured the text is that it allows a reader to focus on a single application, which way be very useful for those interested in applying these approaches.

Comment: I don´t see any reason for the part 3.1 – i.e. Overview of approaches - to be included in the Result, I recommend moving this part to the Introduction.

Response: The reviewer makes another good point. We have moved the overview of methods and Table 1 to the introduction (lines 125-134, 152-158).

Comment: I do not consider it happy when it is necessary to work with references (other published papers) to understand the results, a large part of whichis presented in the Appendices, and when the original published results are referred to in the form of “Fig 3A in the original study...”

Response: We thank the reviewer for this helpful comment. We have extended the descriptions of previous work, so that a little more context is provided in the main manuscript (e.g., lines 318-323, 326-328, 338-342). We have kept Appendix 1 separate from the main results section in the paper, for two reasons. First, we think this part of the study is the most straightforward (i.e., comparisons of empirical GF data for different groups), so to most readers many examples of applications in the main text will not have added value. Second, the description in Appendix A is necessarily long, and adding it to the main results section would make the paper inordinately long. Finally, references to specific figures from other papers are only included so it is entirely clear which data we are referring to. We have removed these references in the results section (lines 338-342 and 344-347), but kept them in Appendix A for completeness (lines 580 and 588).

Comment: The numbering of Results Sections in the text should be corrected (L275, L307, etc.).

Response: Unfortunately, it appears as if the line numbers are not consistent between the version of the PDF provided to the reviewers and the one that we generated, so we are not sure which exact headings the reviewer is referring to. During our revision, we have double checked all heading numbers, as well as figure and table references, and made changes where needed (e.g., Section headings for the Discussion and Methods were wrong).

Reviewer 3 Report

Comments and Suggestions for Authors

In the submitted manuscript, Johnson and Zwart describe quantitative statistical analysis of the ratio of nucleic acid molecules for multipartite and segmented plant viruses through the re-analysis of previously published data sets. The authors describe their statistical approach and analyze three sets of data, showing that for the most part their results are consistent with previous studies. The manuscript is well written and easy to follow. While the results demonstrate some novelty regarding good practices for the analysis of multipartite genomes, this study is slightly redundant as it is based on re-analysis of previously published data. This manuscript is an interesting meta-examination of multiple multipartite viruses, and will be of interest to virologists and evolutionary biologists. While the manuscript is very well presented, there are some minor issues to address. 

Major issues:

1) Description of viral genome organization, line 29-34. This definition and explanation of virus genome strategies is somewhat limited and brief. Further explanation here would be beneficial, especially when compared to the cite Sicard paper. Define segments as nucleic acid molecules, and improve the definition of different types of genome organization. Subgenomic particles and partial/cryptic molecules could affect this analysis no?

line 30 "differently" this meaning is not clear

Additionally, authors describe the "genome formula" but this is not entirely clear, and being more explicit, describing it as the ratio or relative frequency of nucleic acid molecules in the introduction would be beneficial.

2) The manuscript should briefly describe methods of obtaining the raw data (ie qRT-PCR vs NGS), both in the introduction and for each specific study. It might be worthwhile to comment on any issues arising from the the method used to quantify relative genome segment ratios. How are different RNA segments quantified? IN general, more information on the data sets used, particularly in the method's section, and how they were obtained would greatly improve the paper. 

Minor issues:

Line 67 "cause problems" not very descriptive. Cause disease? Economically important viruses?

ln84: This line looks awkward, Change interpretating (not a real word) to interpreting, and remove second "of" to be "We provide a framework for interpreting our metric's values"

ln 272: "we think" is not appropriate. provide a clearer statement that has evidence to support it.

ln 274: remove "as"

ln 291: one major discrepancy - which one? 

Author Response

We thank the reviewers for evaluating the manuscript and for their constructive comments. Below, reviewer comments are reproduced and our responses are given. Line numbers refer to the revised manuscript, in which changes have been highlighted.

Comment: In the submitted manuscript, Johnson and Zwart describe quantitative statistical analysis of the ratio of nucleic acid molecules for multipartite and segmented plant viruses through the re-analysis of previously published data sets. The authors describe their statistical approach and analyze three sets of data, showing that for the most part their results are consistent with previous studies. The manuscript is well written and easy to follow. While the results demonstrate some novelty regarding good practices for the analysis of multipartite genomes, this study is slightly redundant as it is based on re-analysis of previously published data. This manuscript is an interesting meta-examination of multiple multipartite viruses, and will be of interest to virologists and evolutionary biologists. While the manuscript is very well presented, there are some minor issues to address. 

Response: We thank the reviewer for their appreciation of our work and the importance of thorough analyses.

Major issues:

Comment: 1) Description of viral genome organization, line 29-34. This definition and explanation of virus genome strategies is somewhat limited and brief. Further explanation here would be beneficial, especially when compared to the cite Sicard paper. Define segments as nucleic acid molecules, and improve the definition of different types of genome organization. Subgenomic particles and partial/cryptic molecules could affect this analysis no?

line 30 "differently" this meaning is not clear

Response: We thank the reviewer for these helpful suggestions. Defining segments as molecules is clear language, and we have now done so (lines 31-39). We have explained the different organizations better, and included a new illustration (Figure 1, line 96-109) to ensure this point is clear. Other nucleic acid molecules such as satellites and subgenomic segments, indeed are highly relevant, as we now point out in the introduction (lines 64-69).

Comment: Additionally, authors describe the "genome formula" but this is not entirely clear, and being more explicit, describing it as the ratio or relative frequency of nucleic acid molecules in the introduction would be beneficial.

Response: From our perspective, the genome formula is a concept (unequal abundance of genome segments), and both ratios and relative frequencies are legitimate ways to describe it in quantitative terms. We have made the definition more explicit (lines 48-52).

Comment: 2) The manuscript should briefly describe methods of obtaining the raw data (ie qRT-PCR vs NGS), both in the introduction and for each specific study. It might be worthwhile to comment on any issues arising from the the method used to quantify relative genome segment ratios. How are different RNA segments quantified? IN general, more information on the data sets used, particularly in the method's section, and how they were obtained would greatly improve the paper.

Response: We thank the reviewer for this insightful comment. Indeed, our previous work has shown that quantification method can have an effect on the results (reference 20 as cited in the paper). All of the data re-used here were obtained by RT-qPCR or qPCR, with the exception of one study that explicitly considers the effect of method on the quantification of the genome formula (i.e., reference 20). We also referenced another study (reference 30) which has explicitly tested the effect of rolling circle amplification before sequencing on genome formula quantification for ssDNA virus. The mentioned study (reference 30) is not one of the studies included in the re-analysis of data. We now give this information on methodology in the intro (lines 111-124), and provided a brief reminder in the results section (lines 246-250).

Minor issues:

Comment: Line 67 "cause problems" not very descriptive. Cause disease? Economically important viruses?

Response: This sentence has been made more specific (line 81).

Comment: ln84: This line looks awkward, Change interpretating (not a real word) to interpreting, and remove second "of" to be "We provide a framework for interpreting our metric's values"

Response: We thank the reviewer for spotting these mistakes and have corrected them (line 147-148).

Comment: ln 272: "we think" is not appropriate. provide a clearer statement that has evidence to support it.

Response: This sentence has been removed, as it was superfluous in the revised paper (lines 316-337).

Comment: ln 274: remove "as"

Response: This mistake has been corrected (lines 333-335).

Comment: ln 291: one major discrepancy - which one? 

Response: The case which forms a discrepancy has been made explicit (lines 354-356).

Round 2

Reviewer 2 Report

Comments and Suggestions for Authors

The changes made by the authors significantly improve the orientation in the text and its clarity, including the understanding of results and their context. I recommend the manuscript for publication.
I must comment on one minor point, the Materials section should be included between Introduction and Results, according to the journal's custom. But this is only a technical problem